# Otoplasty through Ventral Skin Incision and Shaping of the Antihelix by Abrasion—A Retrospective Study

**DOI:** 10.3390/jcm10163713

**Published:** 2021-08-20

**Authors:** Shafreena Kühn, Nadine Wöhler, Andrej Wehle, Lara Küenzlen, Jens Rothenberger, Robert Sader, Gottfried Lemperle, Ulrich Michael Rieger

**Affiliations:** 1Department of Plastic and Aesthetic, Reconstructive and Hand Surgery, Agaplesion Markus Hospital, 60431 Frankfurt, Germany; lara.kueenzlen@agaplesion.de (L.K.); jens.rothenberger@agaplesion.de (J.R.); lemperle8@aol.com (G.L.); Ulrich.Rieger@agaplesion.de (U.M.R.); 2Department of Orthopedic and Trauma Surgery, Nordwest Hospital, 60488 Frankfurt, Germany; nadine.woehler@web.de; 3Department for Oral, Cranio-Maxillofacial and Facial Plastic Surgery, University Hospital Frankfurt, Goethe University, 60590 Frankfurt, Germany; r.sader@em.uni-frankfurt.de

**Keywords:** otoplasty, ventral otoplasty, Lemperle, protruding ear, antihelix shaping

## Abstract

(1) Background: Protruding ears are the most common auricular malformation affecting approximately 5% of the population. One common factor leading to auricular protrusion is a deficiency or total absence of the antihelix. A technique first described by Gottfried Lemperle in 2003 attempts cartilage thinning, folding, and fixation by non-absorbable mattress sutures after ventral skin incision along the ventral helical rim. (2) Methods: Retrospective analysis of patient records was performed for otoplasties according to this technique, performed between 1985 and 2014 at Agaplesion Markus Hospital in Frankfurt, Germany. All recorded complications were examined. (3) Results: A total of 912 single otoplasties were performed according to this technique from 1985 to 2014. Overall complications included 26% minor complications not requiring further surgery and 11% major complications leading to revision surgery. Within those requiring revision surgery, the most common reason was recurrence of auricular protrusion (5%), followed by suture granulomas (5%) and hematomas (2%). (4) Conclusions: Lemperle’s otoplasty technique addresses the open thinning and shaping of the antihelix through a ventral incision along the helix to prevent irregularities and possible ridges. Results show a low complication rate comparable to data found in published studies. This technique is easy to perform, safe, and avoids often seen contour irregularities of the antihelix compared to techniques with a posterior approach.

## 1. Introduction

A protrusion of the ear represents a physical attribute encountered in approximately 5% of the human population [1]. An aplasia or hypoplasia of the antihelix, a disproportionally enlarged concha, or an increased conchoscaphal angle may result in a protrusion of the ear, which is noticeable and undesirable to the patient. This deformity is the target of numerous corrections through invasive and non-invasive techniques that have been developed and published over time [1,2,3].

Correction of protruding ears is often performed in children and adults suffering from related psychosocial issues [4] and hoping for aesthetic improvement. Preschool children fall in a convenient timeframe for corrective surgery, as this age correlates with reaching full development of the ear. Successful non-surgical correction techniques are available and have shown to be most effective during the first few weeks after birth, when the soft and pliable cartilage can still be molded by external measures [2,5].

The need for the correction of the antihelix in surgical otoplasties has been the focus of multiple publications. One of the pioneers in addressing the antihelix was Stenström [6], who described the need for increasing the ventral area of the flat antihelix by incising or abrasing the ventral perichondrium. Subsequently, the antihelix can be bent easily and fixed with two or three buried mattress sutures. This surgical technique of antihelical folding was the reason for the ventral incision and antihelical moulding described by Lemperle in 2003 [5]. More precisely, Lemperle evaluates the result of an irregular and uneven appearance of cartilage after posterior incisions, which also appeared after blind ventral perichondrial weakening in Stenström`s technique. In his attempt to perfect Stenström`s technique, Lemperle suggested a hidden ventral incision along the helix, thus obtaining maximum view of the ventral antihelical area to be abraded. Instead of rasping or incising, he suggested the use of a dermabrador for cartilage thinning, which resulted in an even ventral surface, thus avoiding edgy and uneven cartilage appearance [5].

## 2. Materials and Methods

This retrospective study analyses 937 patients from our Department of Plastic and Reconstructive Surgery at Agaplesion Markus Hospital in Frankfurt, Germany, who underwent aesthetic otoplasty over a 30-year period, from 1985 to 2014. Data collection and analysis was performed retrospectively from 2015 to 2016. Patient follow-up records ranged from 3 to 24 months with a median follow-up of 12 months.

All subjects (or their parents or guardians) have given their written informed consent, and the study protocol was approved by the institute’s committee on human research. Outcome measures were minor and major complications according to Clavien and Dindo classification [7], as well as the need for revision surgery, including recurrences requiring revision surgery. Data were collected for each ear separately, and all reanonymized data were presented according to outcome measures and visualized using Microsoft Excel 16.0.12527.20278 (Microsoft Corporation, Redmond, WA, USA) and GraphPad Prism 8.4.1 (GraphPad Software, San Diego, CA, USA).

For the otoplasty technique according to Lemperle, the skin is incised ventrally in the scapha along the entire helix, and from there, bluntly lifted with an elevator or pointed scissors across the antihelix to be formed to the concha. Then, the perichondrium is ground off under sight with a diamond grinding head or a metal brush from the dermabrasion set (Figure 1a), whereby the ear is best held over the index finger of the other hand. After having ground off half of the cartilage, it usually folds itself into the desired antihelix. This is followed by the fixation of the folded antihelix with two or three monofilament non-absorbable U-sutures, 4–0 Ethicon Ethilon Polyamid 6 (One Johnson & Johnson Plaza, New Brunswick, NJ, USA) (Figure 1b), where the knot is facing towards the concha (Figure 1c). Sutures should be placed according to individual folding requirements similar to other techniques with knots lying proximal to the concha and away from the helix. Suture placement may be placed for trial before tying the knot. The skin is then pulled over the newly created antihelix and closed. Ventral otoplasty in comparison to other approaches was primarily preferred in our department; however, the ventral approach does not allow for correction of distinct hypertrophy of the concha. In case of the need for conchal reduction, the dorsal approach was preferred. For patients under the age of 18 years, otoplasty usually was performed under general anesthesia. Otherwise, local anesthesia was preferred.

## 3. Results

The population receiving otoplasty surgery during the 30-year period comprised a total of 937 patients; however, 255 patient records were either incomplete or missing. Out of the remaining population of 682 patients, 631 patients received primary uni- or bilateral otoplasties. For the remaining 51 patients, secondary or tertiary otoplasties were performed. Out of the primary otoplasty procedures, a total of 912 single primary otoplasties were performed in a total of 504 patients according to Lemperle’s technique using ventral incision and cartilage moulding, as shown in Figure 2 [5]. This population included 319 female patients, 180 male patients, and 5 patients without disclosure of gender. The median patient age was 12.1 years (range 1.1–63.6 years).

Observed complications in our study population were hematoma, recurrence of ear protrusion, occurrence of suture granulomas, impaired wound healing (including minor wound margin necrosis or dehiscence), hypertrophic scarring, and infection.

Table 1 outlines complications requiring revision surgery, classified as Clavien–Dindo Grade III [7].

The rate of overall revision surgery (11%, *n* = 97) and for each respective complication is outlined in Table 1. These include hematoma (2%, *n* = 19), recurrence of protrusion (5%, *n* = 43), suture granuloma (4%, *n* = 32), impaired wound healing (1%, *n* = 7), and hypertrophic scarring (0.1%, *n* = 1). Overall revision rates are counted per ear (Table 1).

Table 2 outlines minor complications that did not require revision surgery at any point. These may include conservative local treatment or prescription of antibiotics, or they may not have required treatment at all, classified as Clavien–Dindo Grade I [7]. Overall minor complications (26%, *n* = 234) included minor hematoma (1%, *n* = 104), suture granuloma (8%, *n* = 72), impaired wound healing, such as minor local inflammation or swelling or minor disruption of wound closure (4%, *n* = 32), minor infection (2%, *n* = 21), and hypertrophic scarring (0.6%, *n* = 5) (Table 2).

Figure 3 shows an overall outline of major complications (Grade III) requiring revision surgery and minor complications (Grade I) not requiring surgical revision in terms of total population.

## 4. Discussion

While the projection of the ear from an anterior view remains constant throughout life, the auricular width is found to reach its maximum at the age of 6 years [8]. Ear projection has been found to average 20.4 mm with a range of 12 mm to 28 mm [8,9]. Concerning 5% of the pediatric population [1,3,10] the protruding ear is the most common auricular variance, presenting as a hereditary trait [8]. Whilst genes seem to be the main influencing factor, external factors such as hypoxia or radiation during embryological development are also discussed as possible influences in auricular malformation, but they have yet to be linked to ear protrusion in particular [9].

Non-surgical methods such as early splinting have been described and implemented with some degree of success, if applied at an early postpartum stage [3,6,11,12]. Complications in non-invasive methods have been linked to the onset of treatment at a later infant age [11,12]. Since most German health insurances cover the expenses of surgical correction of distinct ear protrusion for preschool aged children, this factor may inhibit the decision for neonatal ear moulding by simple splinting.

When considering surgical action in treating a protruding ear, parents often search consultation for their affected child around preschool age, when the cartilage is still soft. Furthermore, this time of age is appropriate not only from an anatomical point of view but is also a favorable age in terms of psychosocial impact, as children are yet to be enrolled into primary school. Surgical otoplasty for auricular prominence is highly beneficial for children from a psychosocial point of view [13], with a significant decrease in bullying experiences, increase in self-confidence and overall happiness, as well as improved social experiences [13].

The aesthetics of the auricle is of main concern when it comes to tackling the issue of prominent ears, as surgical otoplasty presents a reconstructive measure in correcting a partially aesthetic, yet non-functional, deficiency with profound psychosocial impact on the lives of the affected [8]. Multiple approaches have been established and are widely available in attempts to perfect otoplasty, both from a surgical and an aesthetic point of view [1,3,7,9,10,14,15,16,17,18]. Surgeons have modified established techniques to further advance them for the benefit of the surgeon and patient. Lemperle postulated that through his direct approach and direct visualization, antihelical construction could be performed more easily and precisely, in comparison to dorsal incision techniques, presenting a useful technique especially for the inexperienced surgeon [14,16,19]. Furthermore, Lemperle implied that by using dermabrasion instead of scoring [16,19] uneven, irregular surfaces and sharp edges [8] could be avoided to achieve a more even and harmonious surface and overall appearance [5]. In addition, the thinner the cartilage, the sharper the antihelix, and the more likely it is to break the edges. The thicker the cartilage is left, the greater the risk of recurrence [5]. As this technique primarily promotes antihelical folding, the crura antihelices may likewise be hereby defined. Whilst being especially suitable for antihelical fold deficiencies, other severe ear deformities—e.g., prominent concha auricularis or excessive concha mastoid angle—may not be addressed appropriately by this technique.

A main concern of this approach is a more visual and perhaps obvious scar. Data available for this study do not provide objectification regarding visibility of the scar, presenting a limitation. However, non-empirical experiences in our department usually show no visibility of the ventral scar without everting the helix. In comparison to data found in publications of dorsal incision techniques, scar revision or correction (e.g., due to hypertrophic scaring) was of no higher frequency for our examined population (Figure 3) [20]. Involved surgeons’ subjective evaluation of the scarring being negligible cannot be objectified in our study but was commonly suggested. However, this ventral approach may be used more cautiously and reluctantly in patients known or suspected to suffer from keloid or hypertrophic scaring. Although the literature shows palpable or protruding suture knots to be a relevant complication, even leading to respective usage of flap techniques for knot coverage [20,21], the follow-up of this study has not shown palpable sutures or knots to present a complication of this technique. This may also be because knots placed in the deep conchal cave are less likely to be palpated by the patient, hence being less noticeable.

With regards to complication rates, our data show a comparably low rate of recurrence in ear protrusion (5%; Table 1), as current literature suggests rates between 3% and 12% [22,23] depending on the individual technique applied. Sole skin excision presented the highest recurrence rates, and this was followed by cartilage moulding techniques, then by cartilage breaking techniques, which showed the lowest rates of recurrences [22]. Additionally, one recent study showed posterior cartilage modification according to a modified Mustardé technique, to result in lower recurrence rates (3%) compared to ventral cartilage manipulation (10%) according to a modified Chongchet technique [23]. Hematomas have been described to occur in 1–33% of cases after otoplasty, according to individual techniques and publications [23]. Wound infection was found to be 2% for our study compared to 1% for the modified Mustardé technique [23]. Occurrence of granuloma requiring revision surgery for our study population was 4% (Table 1). Further complications requiring revision surgery were wound-related impairments, such as minor skin necrosis, or wound dehiscence (1%; Table 1). Surgical correction of hypertrophic scarring occurred in 1 ear (0.1%; Table 1, Figure 4). Although minor complication rates seem high, it should be noted that every minor healing disorder (Clavien–Dindo Grade I) was considered, including those not requiring any therapeutic intervention or management strategies at all. Overall, these results show comparable and low rates of complication, both regarding those requiring surgical revision, as well as those treated conservatively or not requiring complication treatment at all. According to our experience, both the learning curve and patient satisfaction are high for this technique, although this study does not provide objectification, and further research is needed.

In general, hypertrophic scars and keloids on ears occur only after local infections [24]. Therefore, it is no wonder that there is only one hypertrophic scar in our series of mainly children, and almost all scars along the inner helix became invisible over the following months. Therefore, we can recommend this technique to be easy, safe, and aesthetically effective, as shown in Figure 5.

## Figures and Tables

**Figure 1 jcm-10-03713-f001:**
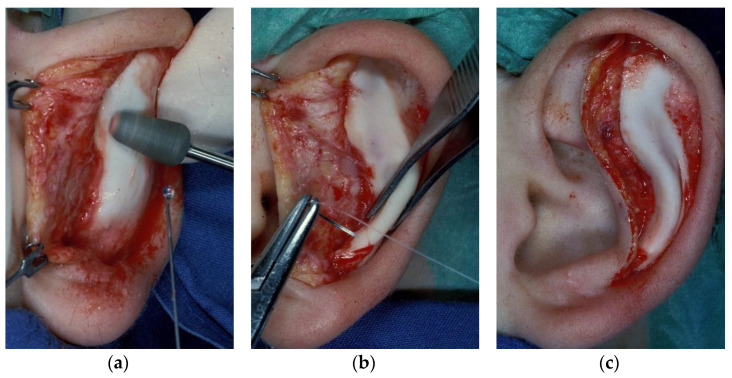
Ventral otoplasty technique. (**a**) Skin incision is shown along the scapha, followed by flap-raising until the area of the anticipated antihelical fold is fully exposed. A diamond grinder is used along the anticipated antihelical fold to abrase and thin out the cartilage. By regular palpation of this area, abrasion is performed until the cartilage shows an adequate thinness for folding. (**b**) U-sutures are placed with knots lying proximal to the concha and away from the helix after cartilage thinning. Two or three knots are placed along the antihelical fold for definite fixation. (**c**) This shows the result after fold fixation. Knots are not seen because sutures are colorless and lie proximal to the conchal area and are disguised by its tissue.

**Figure 2 jcm-10-03713-f002:**
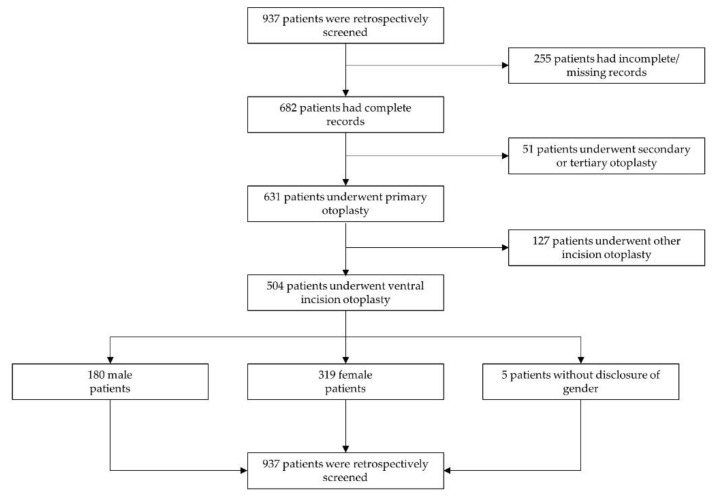
Enrollment of patients. Shows the screening and enrollment of patients.

**Figure 3 jcm-10-03713-f003:**
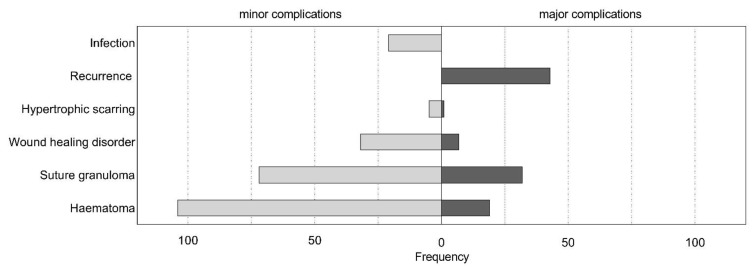
Histogram of individual minor and major complications in direct comparison following the Lemperle otoplasty technique.

**Figure 4 jcm-10-03713-f004:**
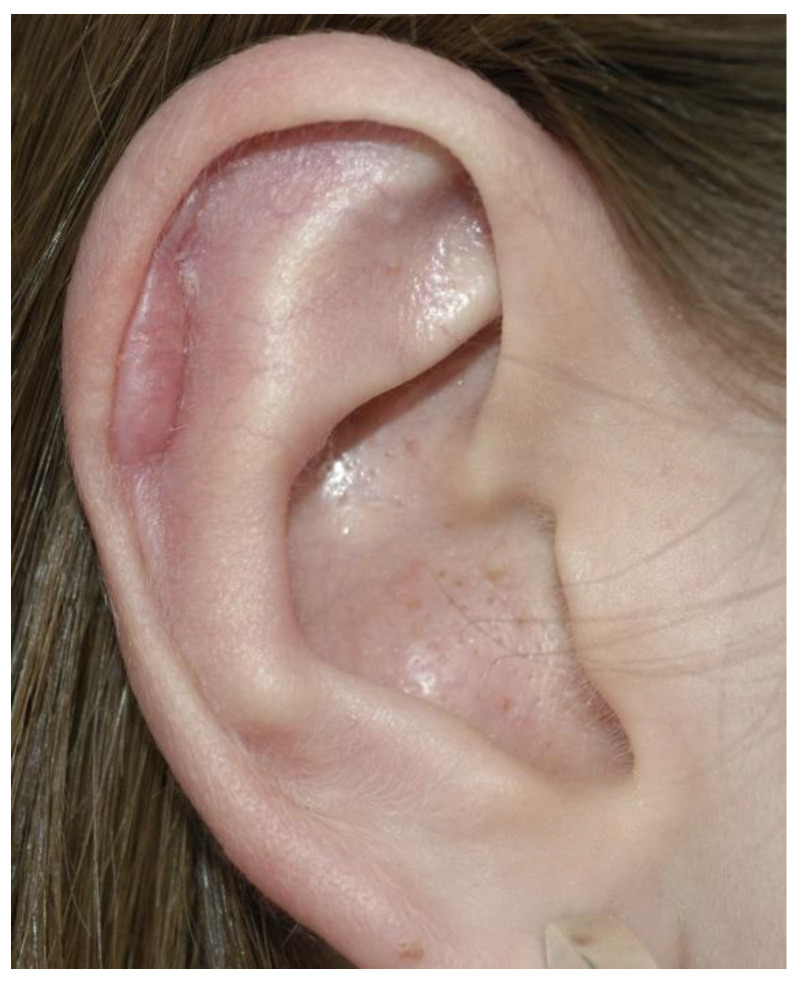
Hypertrophic scar after ventral approach.

**Figure 5 jcm-10-03713-f005:**
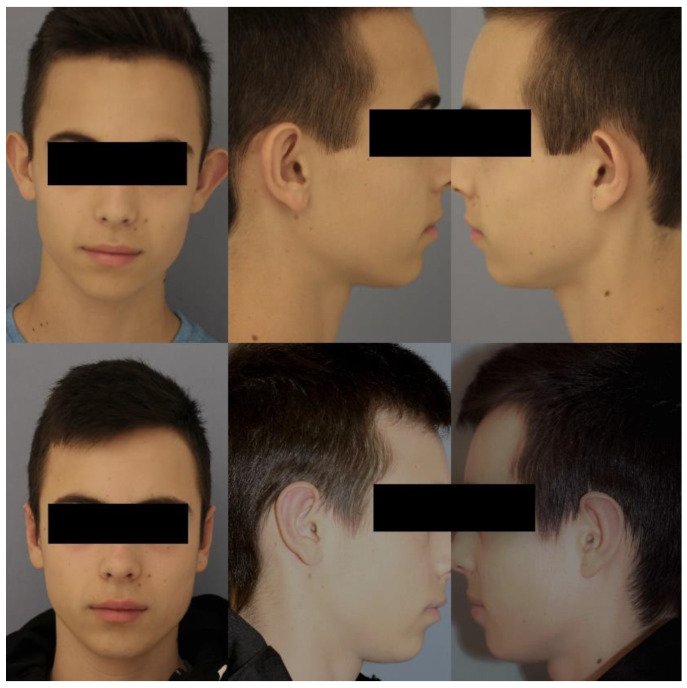
Fifteen-year-old patient with bilateral asymmetrical ear protrusion; the left antihelix shows greater deficiency compared to the right (anterior view, lateral view). The same patient after ventral otoplasty according to Lemperle’s technique; the left antihelix shows greater folding to reach more symmetry. Scars are hidden in the scapha and cannot be seen without manipulation (anterior view, lateral view).

**Table 1 jcm-10-03713-t001:** Major complications requiring revision surgery following primary ventral otoplasty according to the Lemperle technique (*n* = 912).

	Overall Revision Surgery ^1^	Hematoma	Recurrence ^2^	Suture Granuloma	Impaired Wound Healing	Hypertrophic Scarring
All	11% [97/912]	2% [19/912]	5% [43/912]	4% [32/912]	1% [7/912]	0.1% [1/912]
Unilateral	57% [55/97]	58% [11/19]	58% [25/43]	56% [18/32]	100% [7/7]	100% [1/1]
Bilateral	43% [42/97]	42% [8/19]	42% [18/43]	44% [14/32]	0% [0/7]	0% [0/1]

^1^ Revision surgery as per ear. ^2^ Including recurrent ear protrusion and other minor corrective measures for asymmetry, overcorrection, and insufficient antihelical folding.

**Table 2 jcm-10-03713-t002:** Minor complications not requiring revision surgery following primary ventral otoplasty according to Lemperle technique (*n* = 912).

	Overall Minor Complications	Minor Hematoma	Suture Granuloma	Impaired Wound Healing ^1^	Minor Infection	Hypertrophic Scarring
All	26% [234/912]	11% [104/912]	8% [72/912]	4% [32/912]	2% [21/912]	0.6% [5/912]
Unilateral	61% [142/234]	54% [56/104]	58% [42/72]	88% [28/32]	62% [13/21]	60% [3/5]
Bilateral	39% [92/234]]	46% [48/104]	42% [30/72]	12% [4/32]	38% [8/21]	40% [2/5]

^1^ Impaired wound healing included minor superficial wound dehiscence, minor marginal necrosis, or prolonged wound healing not requiring further treatment.

## Data Availability

The data presented in this study are available on request from the corresponding author. The data are not publicly available due to ethical, legal or privacy issues of participants.

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
