# Peer review of "Otoplasty through Ventral Skin Incision and Shaping of the Antihelix by Abrasion—A Retrospective Study"

_jcm, 2021, doi:10.3390/jcm10163713_

Round 1
Reviewer 1 Report
This manuscript presents an extensive review of a lesser known otoplasty technique. Very nicely done. A few suggestions to improve the manuscript:
1) Is there data for the 127 patients who underwent otoplasty by a different incisional approach? If so, this could be an excellent control group. Either way, please describe the decision making process the authors use in deciding whether to use ventral approach or a different approach. I would imagine it involves whether the concha is enlarged but this decision making should be presented in the methods or discussion section.
2) Please include the timing of data collection and analysis in the Methods section.
3) What was the minimum and median follow up of patients?
4) Line 178 (frequency of hypertrophic scarring in ventral technique compared to dorsal approaches) needs a citation.
5) Do the authors typically perform this procedure under local anesthesia only, local with sedation, or general anesthesia?
Author Response
Dear Editor, dear Reviewers,
Thank you for revising our manuscript thoroughly and for your very constructive input and highly valued suggestions. All changes in the manuscript have been highlighted accordingly. We would kindly like to respond in chronological order as follows.
1) Unfortunately data and records regarding the 127 patients who underwent different incisional approaches is incomplete. This is to an extent which does not allow for an statistical approach as a control group.
A describtion regarding the decision making process was added to the manuscript. Ventral approach was prefered if an enlarged concha needed correction.
2) Information about data collection and analysis was included. Retospective data was collected from patientens undergoing otoplasty from 1985-2014. Data collection and analysis was performed retrospectively from 2015 to 2016.
3) Information about minimum and median follow-up was added.
4) A citation was added.
5) Information about anesthesia was included. For patients under the age of 18 otoplasty is usually performed under general anesthesia.
We sincerely hope to have met your criteria and the changes we made to the manuscript are of satisfaction to you. Thank you again for your efforts and time invested in revising and improving this manuscript. We highly value your opinion and your work and feel honored to be considered for publication by you.
Sincerely,
Shafreena Kühn et al.
Reviewer 2 Report
discussion section, line 135: 20.4 mm?
discussion section, line 174: this is the central point of the paper and why most people go post auricular.
Author Response
Dear Editor, dear Reviewers,
Thank you for revising our manuscript thoroughly and for your very constructive input and highly valued suggestions. All changes in the manuscript have been highlighted accordingly. We would kindly like to respond in chronological order as follows.
1) Line 135 was corrected.
2) Unfortunately records and data did not allow for a evaluation of visibility of the scar, however from the authors experience scars are usually not visible at all after ventral approach. We added this information about our personal non-empirical experience with scars after ventral approach to the manuscript.
We sincerely hope to have met your criteria and the changes we made to the manuscript are of satisfaction to you. Thank you again for your efforts and time invested in revising and improving this manuscript. We highly value your opinion and your work and feel honored to be considered for publication by you.
Sincerely,
Shafreena Kühn et al.